# A novel application of neural networks to identify potentially effective combinations of biologic factors for enhancement of bone fusion/repair

**Albert T. Anastasio**[1]☯*, **Bailey S. Zinger**[2]☯, **Thomas J. Anastasio**[3]☯

**1** Department of Orthopaedic Surgery, Duke University Medical Center, Durham, North Carolina, United States of America, **2** Chemical and Biological Engineering Department, University of Colorado at Boulder, Boulder, Colorado, United States of America, **3** Department of Molecular and Integrative Physiology, University of Illinois at Urbana-Champaign, Urbana, Illinois, United States of America

☯ These authors contributed equally to this work.
* albert.anastasio@duke.edu

**Data Availability Statement:** Data are available at https://github.com/SigmoidNetmaker/deep-network-orthobiologics.

## Abstract

### Introduction

The use of biologic adjuvants (orthobiologics) is becoming commonplace in orthopaedic surgery. Among other applications, biologics are often added to enhance fusion rates in spinal surgery and to promote bone healing in complex fracture patterns. Generally, orthopaedic surgeons use only one biomolecular agent (ie allograft with embedded bone morphogenic protein-2) rather than several agents acting in concert. Bone fusion, however, is a highly multifactorial process and it likely could be more effectively enhanced using biologic factors in combination, acting synergistically. We used artificial neural networks, trained via machine learning on experimental data on orthobiologic interventions and their outcomes, to identify combinations of orthobiologic factors that potentially would be more effective than single agents. This use of machine learning applied to orthobiologic interventions is unprecedented.

### Methods

Available data on the outcomes associated with various orthopaedic biologic agents, electrical stimulation, and pulsed ultrasound were curated from the literature and assembled into a form suitable for machine learning. The best among many different types of neural networks was chosen for its ability to generalize over this dataset, and that network was used to make predictions concerning the expected efficacy of 2400 medically feasible combinations of 9 different agents and treatments.

### Results

The most effective combinations were high in the bone-morphogenic proteins (BMP) 2 and 7 (BMP2, 15mg; BMP7, 5mg), and in osteogenin (150ug). In some of the most effective combinations, electrical stimulation could substitute for osteogenin. Some other effective

**Funding:** The authors received no specific funding for this work.

**Competing interests:** The authors have declared that no competing interests exist.

combinations also included bone marrow aspirate concentrate. BMP2 and BMP7 appear to have the strongest pairwise linkage of the factors analyzed in this study.

## Conclusions

Artificial neural networks are powerful forms of artificial intelligence that can be applied readily in the orthopaedic domain, but neural network predictions improve along with the amount of data available to train them. This study provides a starting point from which networks trained on future, expanded datasets can be developed. Yet even this initial model makes specific predictions concerning potentially effective combinatorial therapeutics that should be verified experimentally. Furthermore, our analysis provides an avenue for further research into the basic science of bone healing by demonstrating agents that appear to be linked in function.

## Introduction

Bone repair is a highly multifactorial process involving a wide array of molecular and cellular factors [1]. Orthopaedic surgeons have manipulated these factors by administering various biologic agents (referred to as "orthobiologics") in order to augment bone repair [2]. In most cases, surgeons have administered only one biologic agent. Considering the physiological complexity of the process, it is reasonable to suggest that superior bone repair could be achieved using biologic factors in combination.

Combinatorial explosion prohibits exhaustive experimental evaluation of the full set of possible combinations. An alternative is to use computational methods to extrapolate, or *generalize*, from existing data and predict which combinations would be the most effective, and then expend experimental resources to evaluate only those. Different kinds of artificial intelligence (AI) could be used for this purpose, but the most powerful AIs in use today for making accurate predictions are artificial neural networks (ANNs), trained on datasets using machine learning (ML) [3, 4]. The goal of this study is to use ANNs to extrapolate from existing, experimental data to identify potentially effective combinations of orthobiologic agents. The use of ANNs in this context is unprecedented. Our study is based on standard, canonical ANNs and ML, which renders transparent the application of these forms of AI to orthobiologic data. Our study provides a framework in which to incorporate new data as they become available in the emerging, but still relatively small, field of orthobiologic research. Yet even this incipient computational foray yields potential new perspectives into possible synergisms between orthobiologic factors that should inspire further experimental work.

ANNs are composed of many, highly interconnected neuron-like elements known as units, which can be arranged in layers or circuits. ANNs are computational devices that process information from their input units to produce a pattern of activation at their output units. Feedforward networks have their units arranged in layers. The simplest feedforward networks have only two layers of units: input and output. More complex feedforward networks have one or more layers of hidden units, so called because they are interposed between the input and output layers. Feedforward ANNs are considered deep if they have more than two hidden layers [5]. Recurrent networks have their units arranged in circuits. Multiple processing layers, or circuits, are needed when the production of useful output patterns requires the processing of complex interactions among the inputs.

ANNS are trained via ML on a set of input/desired-output examples. They have been applied in many domains of biomedicine [6–8]. The most extensive medical applications of ANNs have been in radiology [9–12]. Generally in these applications the inputs are the pixels of (usually MRI) images, and the desired outputs are the components of known radiological diagnoses. Once trained, the ANN could generalize from its training data and could make a diagnosis from an image on which it has not been trained. It is likely that clinicians will soon use ANNs adjunctively in radiological diagnosis.

The application of ML has grown rapidly in orthopedics [13–16]. Recent projects have used ML to train not only ANNs but a wide variety of statistical and other modeling modalities as well. Datasets were constructed most often from radiological images (eg X-radiology, computed tomography, magnetic resonance, bone density, etc) but also from patient records, gate patterns, various sensors (eg force sensors, kinetic skeletal trackers, video, wearable device records, etc), and other kinds of measurements (eg biochemical, biomechanical, and bioelectrical). The learning acquired by ML was used for diagnosis of such disorders as osteo-arthritis, ligament deficiency, cartilage lesion, rotator cuff pathology, spinal stenosis, scoliosis, and detection of weakness, age, fracture, tumor, and other bone abnormalities. The learning could also be used to predict surgical outcomes including survival, risk, complications, morbidity, mortality, treatment cost, length of hospital stay, and even patient satisfaction. In orthopedic surgery, knowledge gained via ML was used to detect and identify surgical landmarks or targets such as specific bones, ligaments, and other skeletal components. The application of various biologic agents to bone repair is a rapidly growing subfield of orthopedics. Over the past few decades, many reports have demonstrated the benefits of specific orthobiologic agents on post-surgical bone fusion rates. Yet AI has not been applied in this domain. We apply ANNs trained via ML in this rapidly growing area of orthopedic research.

The usefulness of an ANN derives from its ability to *generalize* beyond its training data, so that it can predict the correct output for inputs on which it has not been trained. We collected a large amount of the available orthobiologic data, organized it into a form suitable for ML, and used it to train an ANN with an architecture that we had determined beforehand would generalize well over the dataset. We used this ANN to explore potential combinatorial therapies within the realm of orthobiologic adjuvants.

## Methods

Our study design consisted of six steps: (1) assemble a dataset on the outcomes associated with the use of various agents and organize them into a form suitable for ML; (2) build a series of ANNs with increasingly complex architectures and processing potentials; (3) determine for each network type its optimal ML parameters; (4) assess the ability of each network type to generalize over the dataset; (5) train the best generalizing ANN on the dataset and use it to predict the efficacy of a large number of combinations of the factors on which it had been trained; and (6) analyze the predictions, to determine which combinations of factors are potentially the most effective and should be experimentally verified. The first five steps are methodological and are summarized here in the Methods section. Further methodological details are available in Supplementary Texts S1–S3 Texts. Step (6) is elaborated in the Results section. All computer programs were written in-house in MATLAB™ and run on Intel Core-i5 based Dell™ workstations. Runtimes (see below) pertain to this software and hardware configuration. Algorithms were developed directly from their descriptions in the primary literature, as summarized and cited in S2 Text. The core set of programs and the entire dataset are available for free download at https://github.com/SigmoidNetmaker/deep-network-orthobiologics.

*Step 1*. We curated the dataset from the experimental literature. We searched for data on the outcomes for bone healing of administration of biologic and other factors. We searched the MEDLINE database using the PubMed search engine. To access basic science and human studies regarding several orthobiologics and other adjunctive therapies for bone healing, we searched the following terms: low intensity pulsed ultrasound, electrical stimulation, rhBMP2, rhBMP7, zoledronic acid, osteogenin, platelet-derived growth factor, bone marrow aspirate concentrate, and platelet rich plasma. Osteoconductive delivery agents were recorded as inputs, as were factors related to study design, such as human or animal model. Dosing, when recorded in a systematic fashion in the study, was also included as an input variable. Data was curated to include adequate experimental sample sizes for each adjunct we ultimately included as an input. Our dataset contains 225 entries, each a specific experimental result (see also *Steps 3* and *4*). Further details on our literature search and citations to the literature we used to compile the training data can be found in S1 Text.

We organized the data into input/desired-output pairs. In total, 17 factors (active orthobiologic agents and their vehicles of administration, or other nonpharmacological treatment types) constituted the inputs, and 26 outcomes (metrics quantifying the efficacy for improvement of bone healing due to the agents) constituted the desired outputs. The inputs and desired outputs are quantified precisely in the dataset, in appropriate units as reported in the literature. We used as many factors as inputs, and outcomes as desired outputs, as were available in the literature, in order to maximize the amount of ANN training data. The entire dataset, along with units and measures as reported in the literature, is provided in the S1 Dataset.

*Step 2*. We constructed a set of 16 different ANN types (see S2 Text for a brief overview of ANNs). Our 8 basic types were feedforward with 0, 1, 2, 3, 5, 7, or 10 hidden layers, and a recurrent network with a hidden circuit (see S3 Text). All hidden layers (and the hidden circuit) were composed of 100 units. We evaluated each of these 8 network types with and without an autoencoder layer. An autoencoder is the hidden-unit representation developed by a network that learns to reproduce its own input at the output. We found that an autoencoder with 50 hidden units provided the best generalizability over our dataset (see S3 Text). Placing an autoencoder layer after the input layer can improve the generalizability of an ANN.

*Step 3*. We optimized the parameters of the ML algorithm used to train each of the ANN types, focusing on the 2 key ML parameters (learning rate and batch size) pertinent to backpropagation and its variants (see S3 Text). This involved assessing network generalizability over ranges of the 2 parameters and choosing the parameters associated with the best generalizability. All ANNs were trained for 50,000 training iterations using backpropagation (or a variant, specifically delta rule or recurrent backpropagation) where each iteration constituted one input/desired-output presentation (see S3 Text). Training for 50,000 iterations took about 15 minutes. In assessing the ability of a network type, trained using specific values for the 2 ML parameters, to generalize, we divided the set of 225 input/desired-output patterns into 175 training patterns and 50 testing patterns. We computed the root mean square (RMS) error over the 50 testing patterns after training on the 175 training patterns. We then retrained the same network type, using those specific ML parameters, 10 times, choosing a different training set and testing set at random each time. The optimal ML parameter value was the one associated with the lowest average RMS error over the 10 retrainings (see also S2 and S3 Texts). The 10 retrainings required about 2.5 hours on average. We tested each network type over a range of about 10 values for each of the 2 ML parameters. In our context, the 2 ML parameters (learning rate and batch size) were independent of one another (ie they did not interact), so iterating on those parameters was not necessary. Nevertheless, finding optimal values for the 2 ML parameters for any single network type required about 2 days of computing time. Finding

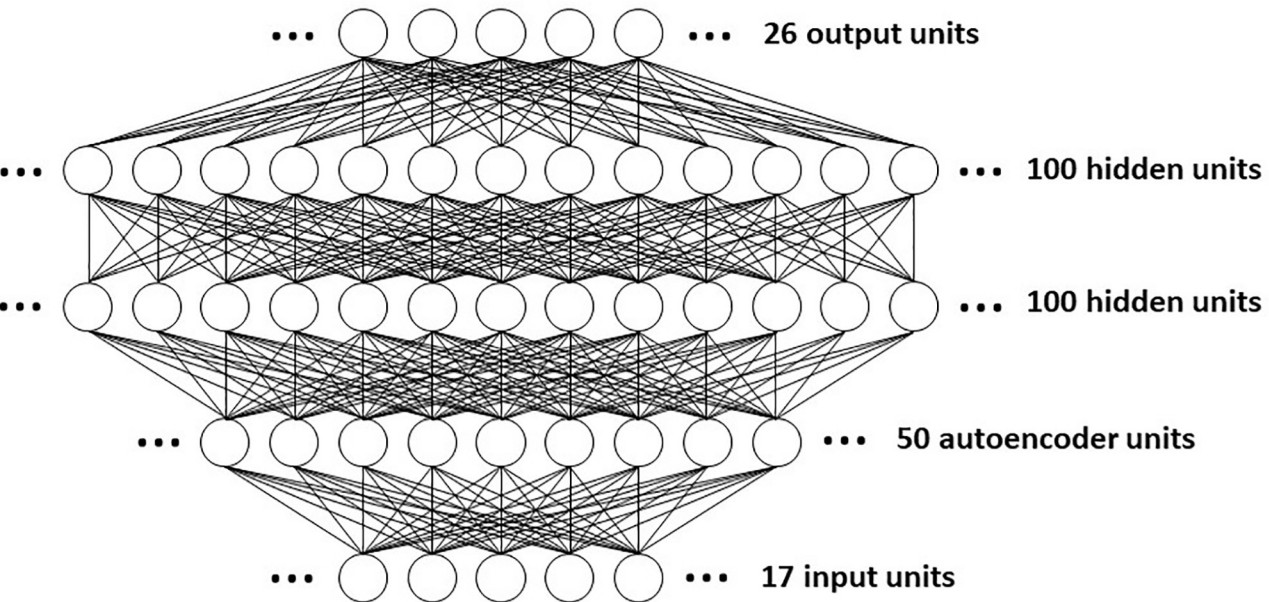

**Fig 1. Diagram of the artificial neural network (ANN) used to predict the efficacy of 2400 combinations of orthobiologic factors.** The feedforward network has two layers of hidden units. The first hidden layer receives the input not directly but only after it has been processed by a layer of autoencoder units. The 17 input units project to a layer of 50 autoencoder units, which project to the first layer of 100 hidden units, which in turn project to the second layer of 100 hidden units, which finally project to the set of 26 output units. The connectivity between layers is complete in that each unit in a previous layer projects to every unit in a subsequent layer. The weights from the input layer to the autoencoder layer are trained separately, and then held fixed while the other weights in the network are trained.

optimal values for the 2 ML parameters (learning rate and batch size) for all of the 16 network types took about 32 days of computing time.

*Step 4*. Having determined the optimal ML parameters for each network type, we then assessed the ability of each network type to generalize. For each network type, each with its optimal ML parameters as determined in *Step 3*, we divided the set of 225 input/desired-output patterns into 175 training patterns and 50 testing patterns, trained for 50,000 training iterations on the 175 training patterns, and computed RMS error over the 50 testing patterns. We repeated this procedure 10 times, choosing a different training set and testing set at random each time. Again, this required about 2.5 hours for each ANN type. The best generalizing network type was the one having the lowest average RMS error over the 10 retrainings (see also S2 and S3 Texts). We found that the feedforward ANN composed of an input layer, an autoencoder layer, two hidden layers, and an output layer exhibited the best generalizability. A diagram of this ANN is shown in Fig 1.

*Step 5*. The full set of 225 input/desired-output patterns was used to train the best generalizing ANN type (Fig 1). Again, this required about 15 minutes of computing time. Following ANN training, we used clinical judgement in deciding which combinations of factors to evaluate, and which outcomes to use in assessing the predicted post-surgical benefit of those selected combinations. We generated a set of 2400 combinations of 9 of the factors that were included among the 17 inputs in the dataset. These factors were chosen because they could be combined appropriately in a surgical setting. The 9 chosen agents were bone-morphogenic protein-2 (BMP2), bone-morphogenic protein-7 (BMP7), osteogenin (OG), platelet-derived growth factor (PDGF), bone marrow aspirate concentrate (BMAC), and platelet rich plasma (PRP) [2]. The vehicles carrying these agents varied greatly among published studies, so we included the most common one, exogenous bone graft (EBG), as a stand-in for all vehicles. Pulsed

ultrasound (US) and electric stimulation (ES) were also chosen for the combination screen because they have been shown to increase bone healing rates [17, 18].

We quantized input levels in order to generate a finite number of input combinations. For the combination screen the factor BMP7 takes 2 levels; BMP2, OG, and PU each take 4 levels; and PDGF takes 5 levels in their ranges. The factors ES, BMAC, and PRP are either present or absent. EBG, as the common vehicle of administration, is present in all combinations.

We further constrained the number of combinations for reasons of practicality. Although PU and ES could theoretically be used in combination, the feasibility of carrying out this dual therapy in practice is low. Due to the additional operative time of harvesting PRP and BMAC to use at the bone healing site, we decided to not to include combinations involving both agents. Editing according to these constraints left 2400 combinations.

To screen the combinations for efficacy, we set the input in turn to each one of the 2400 combinations: the 9 input units corresponding to BMP2, BMP7, OG, PDGF, PU, ES, BMAC, PRP, and EBG took their values as specified for that combination; the other 8 of the 17 input units took value 0. We then computed the activities in response to each input of the 26 output units. Due to randomness inherent in the ML (ie backpropagation-based) algorithms we used, ANNs of the same type trained on the same dataset can nevertheless vary slightly. Therefore, the best predictions are derived from the averaged outputs of several ANNs [19]. We based our predictions on the averaged outputs of 10 separately trained ANNs of the type shown in Fig 1.

To compute a relative efficacy measure for each factor combination, we combined 17 of the 26 averaged output unit activations into a single number. We chose these 17 outcomes because they best assessed the degree of bone healing and functional outcome across studies. The chosen outcomes are distraction rate (DR), bone formation at 3 months (BF3), bone formation at 6 months (BF6), mineralized tissue volume/total tissue volume (MV/TV), 1-level posterior lumbar fusion rate (PLF-FR), Oswestry disability index improvement (ODI), fusion rate (FR), fracture healing percentage (FH), Oswestry Score (OW), radiographic outcome (RO), histomorphometric outcome (HO), implant survival percentage (IS), time to achieve full weight bearing/clinical healing (TWB/CH), mean time to radiographic union (TRU), need for repeat bone grafting (RBG), not healed at end of trial (NH), and need for dynamization (DY). We flipped the outputs whose high score indicated poor efficacy, normalized all outputs into the range [0, 1] and then averaged the 17 normalized outputs to arrive at a single-number efficacy score. By this relative measure, perfectly effective and ineffective combinations would have efficacy scores of 1 and 0, respectively.

## Results

We trained the ANN with the best generalizability (Fig 1) to achieve a good but not perfect match between its actual and desired outputs, because the overtraining required to achieve a perfect match would impair its ability to generalize. Comparison of the desired and actual output images for an example ANN (Fig 2) shows that the agreement is good but not perfect. Precisely this sort of relationship would be expected for an ANN that could generalize beyond its training data.

We rank-ordered the predicted efficacy scores for the 2400 combinations (Fig 3). They ranged from about 0.30 to almost 0.75 and so covered almost half of the possible [0, 1] range. The efficacy scores seemed to plateau for the most effective several hundred combinations.

The 2400 rank-ordered combinations are shown in two separate images in Fig 4: one for all 2400 combinations and another for the top 200. Analysis of the top 200 reveals some statistically significant, pairwise correlations among the 9 factors (Table 1). BMP2 and BMP7 are positively correlated, while OG and PDGF are negatively correlated. PDGF is negatively or positively correlated with BMAC or PRP, respectively. PU and ES, and likewise BMAC and

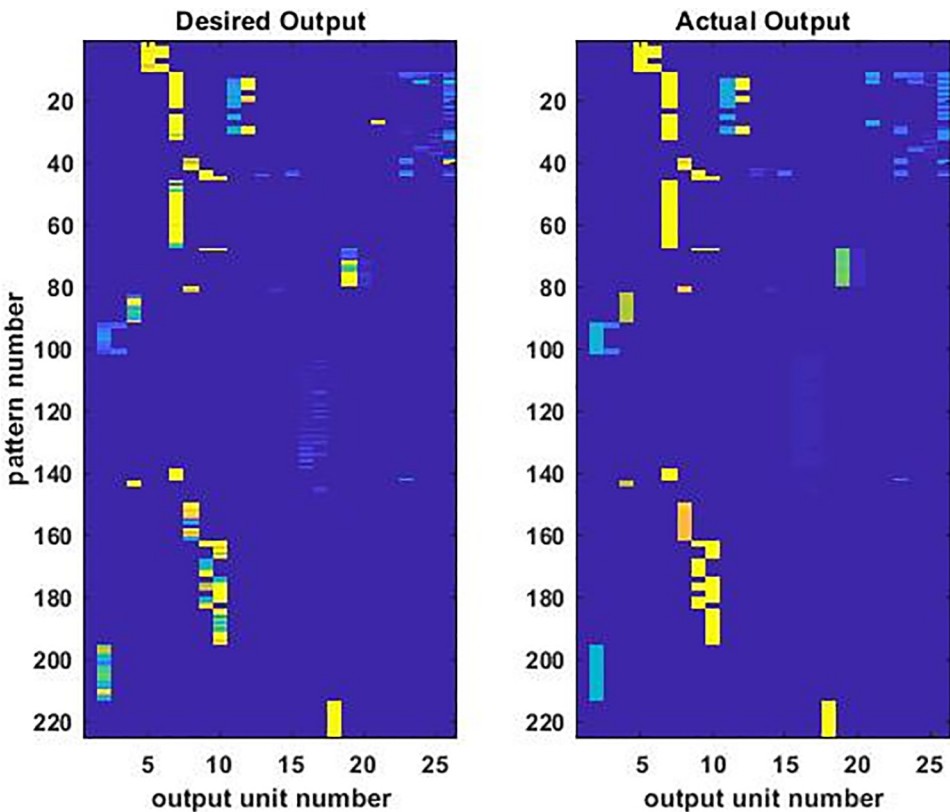

**Fig 2. Desired and actual outputs of the best-generalizing ANN after training.** The values of the 26 output units for each of the 225 input/desired-output patterns in the training dataset are shown as images, separately for the desired (left) and actual (right) outputs. The unscaled values of the outputs range from 2 (deep blue) to 277 (bright yellow). Close inspection reveals that the match between the desired and actual outputs is good but not perfect. This is expected due to ambiguity in the training dataset, which is derived directly from the experimental data of multiple labs that often reported different outputs for the same inputs. The pattern of agreement in general, with disagreement in detail, indicates that the neural network has learned to generalize from the data in the training dataset. The ability to generalize is central to the ability of an ANN to predict the outputs for combinations of inputs on which it has not been trained.

PRP, are also negatively correlated, but this is due largely to constraints in the design of the combination screen (see Methods).

The 10 best factor combinations show some consistent similarities, and some consistent differences with the 10 worst combinations. BMP2 and BMP7 are at their highest levels in the 10 best combinations, while they are 0 in the 10 worst combinations, and this is consistent with the positive correlation observed between BMP2 and BMP7 in the top 200 combinations. OG tends to be at its highest levels in the 10 best combinations but is 0 in the 10 worst combinations. In contrast, PDGF tends to be at its lowest or highest levels in the 10 best or worst combinations, respectively, and this is consistent with the negative correlation observed between OG and PDGF in the top 200 combinations.

The analysis suggests that the most effective combinations are high in BMP2, BMP7, and OG, but low in PDGF (see Table 2 and its caption for quantification of amounts). These 4 factors seem to be the most determinative of the best factor combinations. BMAC appears in some of the 10 best combinations but in none of the 10 worst, and this is consistent with the negative correlation between PDGF and BMAC. PRP is absent from all 10 best and 10 worst combinations. The 10 best and 10 worst combinations seem indifferent to the levels of PU and

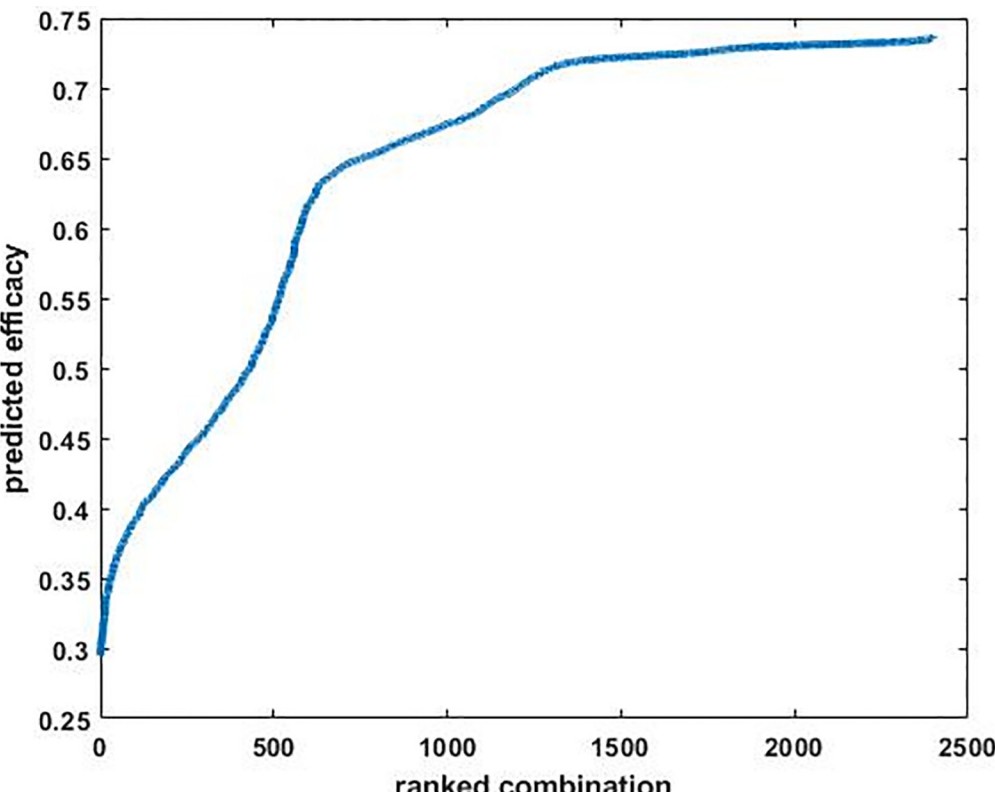

**Fig 3. The efficacies of 2400 combinations of orthobiologic factors as predicted by the best-generalizing ANN.** The scores are sorted from least to most effective. Effective combinations were defined as those that were high in 12 / 26 outputs (DR, BF3, BF6, MV/TV, PLF-FR, ODI, FR, FH, OW, RO, HO, and IS), and low in 5 / 26 outputs (TWC/CH, TRU, RBG, NH, and DY). The remaining 9 / 26 outputs were not included in the efficacy measure (see text). For the purposes of ranking, the outputs that should be low were flipped, all outputs were scaled in the range [0, 1], and the output values were averaged. By this measure, which is relative to the maximal and minimal output values, the highest possible efficacy of 1 would be obtained if all of the outputs that should be high / low were at their maximal / minimal levels for that output, and vice-versa for the lowest possible efficacy of 0. The predicted efficacies for the 2400 combinations in the screen varied widely over the [0, 1] range and nearly plateaued for the most effective several hundred.

ES, with the potentially important exception that ES appears in some of the 10 best combinations that lack OG.

The 2400 combinations in the screen include the null combination (ie none of the 9 factors are present except for EGB, the common vehicle), and all combinations in which EGB and 1 other factor only are present. The analysis clearly indicates that combinations of several orthobiologic factors would be more effective than any single factor alone. The analysis indicates that combinations of BMP2 (15mg), BMP7 (5mg), and OG (150ug), each at the high end of their ranges as shown in the input/desired-output table, and perhaps including ES or BMAC (see S1 Text and S1 Dataset), should outperform combinations that lack those components. Experimental verification of these predictions could lead to the development of orthobiologic factor combinations that outperform single factors for the enhancement of bone repair.

## Discussion

To properly situate our model within the orthopaedic literature, it is necessary to distinguish between process-driven and data-driven models. Process-driven models represent processes

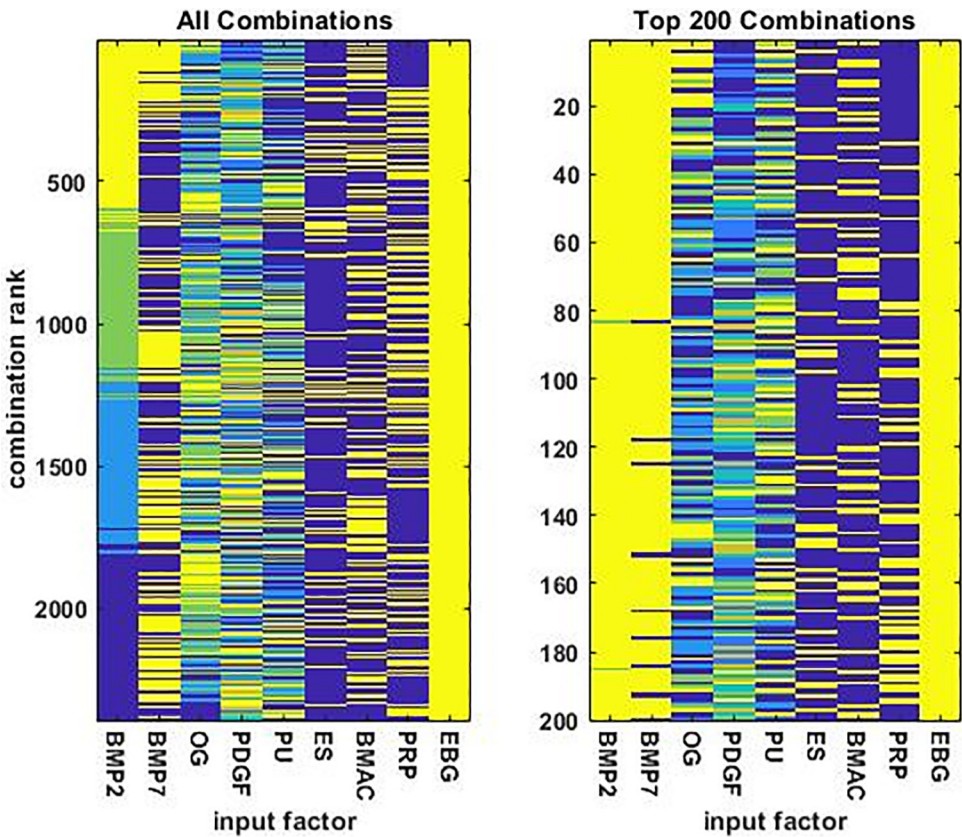

**Fig 4. The combinations of orthobiologic input factors ordered by predicted efficacy.** The combinations are presented in descending order, so the combinations are most / least effective at the top / bottom of either plot. The plot on the left shows all 2400 combinations while the plot on the right shows the top 200 combinations. The factors ES, BMAC, and PRP are either present or absent; EBG is always present. The factor BMP7 takes 2 levels; BMP2, OG, and PU each take 4 levels; and PDGF takes 5 levels. Input factor levels varied over a broad range but were normalized into [0, 1] for purposes of illustration. In the images, yellow and deep blue correspond to 1 and 0, respectively.

**Table 1. Statistically significant pairwise correlations among the factors in the top 200 orthobiologics combinations as determined from the neural network.** BMP2 and BMP7 are positively correlated, while OG and PDGF are negatively correlated. PDGF is negatively or positively correlated with BMAC or PRP, respectively. PU and ES are negatively correlated, and BMAC and PRP are also negatively correlated. The very low (but still > 0) p values associated with those last two negative correlations is attributed mainly to constraints in the design of the combination screen by which PU and ES could not be nonzero together, and BMAC and PRP likewise could not be nonzero together. All values including p values are reported to four significant places.

| Significant Pairwise Correlations among Factors in Top 200 Orthobiologics Combinations | | | |
|---|---|---|---|
| factor 1 | factor 2 | correlation coefficient | p value |
| BMP2 | BMP7 | 0.1962 | 0.0054 |
| OG | PDGF | −0.2538 | 0.0003 |
| PDGF | BMAC | −0.2291 | 0.0011 |
| PDGF | PRP | 0.2265 | 0.0013 |
| PU | ES | −0.5444 | 0.0000 |
| BMAC | PRP | −0.3690 | 0.0000 |

**Table 2. The 10 best and 10 worst combinations of orthobiologic factors, as determined from the best-generalizing ANN (Fig 1).** Combinations are ranked, best to worst, out of the total of 2400 combinations in the computational screen. BMP2 and BMP7 are at their highest levels in the 10 best combinations, while they are 0 in the 10 worst combinations. OG tends to be at its highest or lowest levels in the 10 best or worst, respectively, while PDGF tends in the opposite direction. The 10 best and worst combinations seem indifferent to the level of PU. ES or BMAC is present in some of the 10 best but in none of the 10 worst. PRP is absent from all 10 best and worst combinations. EGB is present in them all but it is present in all 2400 combinations by design and is included only for completeness. Units: BMP2 and BMP7 are in milligrams, OG and PDGF are in micrograms, and PU is in total treatment days; the other inputs are either present or absent. The levels (dosages, intensities, amounts, etc) of all inputs are in the ranges as reported in published studies (see S1 Text and S1 Dataset).

| Ten Best Orthobiologics Combinations | | | | | | | | |
|---|---|---|---|---|---|---|---|---|
| rank | BMP2 | BMP7 | OG | PDGF | PU | ES | BMAC | PRP | EBG |
| 1 | 15 | 5 | 150 | 0 | 150 | 0 | 1 | 0 | 1 |
| 2 | 15 | 5 | 150 | 0 | 100 | 0 | 1 | 0 | 1 |
| 3 | 15 | 5 | 150 | 0 | 150 | 0 | 0 | 0 | 1 |
| 4 | 15 | 5 | 0 | 0 | 0 | 1 | 0 | 0 | 1 |
| 5 | 15 | 5 | 150 | 50 | 150 | 0 | 0 | 0 | 1 |
| 6 | 15 | 5 | 150 | 0 | 100 | 0 | 0 | 0 | 1 |
| 7 | 15 | 5 | 150 | 0 | 50 | 0 | 1 | 0 | 1 |
| 8 | 15 | 5 | 150 | 50 | 150 | 0 | 1 | 0 | 1 |
| 9 | 15 | 5 | 0 | 0 | 0 | 0 | 0 | 0 | 1 |
| 10 | 15 | 5 | 0 | 50 | 0 | 1 | 0 | 0 | 1 |

| Ten Worst Orthobiologics Combinations | | | | | | | | |
|---|---|---|---|---|---|---|---|---|
| rank | BMP2 | BMP7 | OG | PDGF | PU | ES | BMAC | PRP | EBG |
| 2391 | 0 | 0 | 0 | 100 | 100 | 0 | 0 | 0 | 1 |
| 2392 | 0 | 0 | 0 | 100 | 50 | 0 | 0 | 0 | 1 |
| 2393 | 0 | 0 | 0 | 150 | 150 | 0 | 0 | 0 | 1 |
| 2394 | 0 | 0 | 0 | 150 | 100 | 0 | 0 | 0 | 1 |
| 2395 | 0 | 0 | 0 | 150 | 0 | 0 | 0 | 0 | 1 |
| 2396 | 0 | 0 | 0 | 200 | 150 | 0 | 0 | 0 | 1 |
| 2397 | 0 | 0 | 0 | 150 | 50 | 0 | 0 | 0 | 1 |
| 2398 | 0 | 0 | 0 | 200 | 100 | 0 | 0 | 0 | 1 |
| 2399 | 0 | 0 | 0 | 200 | 50 | 0 | 0 | 0 | 1 |
| 2400 | 0 | 0 | 0 | 200 | 0 | 0 | 0 | 0 | 1 |

explicitly. There is a long tradition of process-driven modeling in bone fracture healing (see [1, 16] for review). Process-driven models are valuable in that they explicitly describe the processes involved, but they are limited to what is known about the processes themselves. This limits their predictive power.

Data-driven models are built almost entirely on observed input-output relationships, without regard for the specifics of the underlying processes. Data-driven models offer little mechanistic insight, but they provide a powerful means to leverage all available data for predicting the outputs to novel inputs. Deep neural networks are the premier form of data-driven modeling in AI today. The multilayered ANN we chose to make our predictions (Fig 1) is a deep neural network [5]. To our knowledge, our model is the first data-driven, deep neural network model of the relationship between biologic factors and bone repair.

Even though it is data-driven, our model may indicate avenues for further research into the molecular physiology of bone healing. For example, OG (osteogenin, or bone morphogenic protein-3 (BMP3)) and ES (electric stimulation) seem to act interchangeably in our model. Interestingly, ES has been shown to upregulate BMPs 2 through 8, and is effective in upregulating BMP3 (also called OG) in cultured bone cells [20]. The fact that our model is able to post-dict previously known molecular pathways advocates for its use in predicting previously unknown molecular interactions.

The main limitation in our study was in the size and composition of the dataset on which we trained our ANN. At 225 input/desired-output training patterns, our dataset is large in comparison with other datasets that are curated from the literature but still very small in comparison with datasets used to train ANNs in many applications. Also, most of the input/desired-output patterns in our dataset included only one active orthobiologic factor. The risk in training mainly on single factors is that the network would fail to learn interactions among them but in our case, it seems that this did not occur.

If ML failed to pick up interactions, then the simplest ANN, that composed only of input and output layers, would have generalized as well as, if not better than, ANNs with hidden layers (or circuits) intervening between input and output (see S4 Text). The fact that the ANN that generalized best over our dataset was a multilayered network strongly suggests that it did learn some of the interactions between the factors.

The best way to remedy the main limitation in this analysis is to train deep neural networks on larger datasets containing more combinations of factors. The analysis already suggests both good and bad combinations that could be explored experimentally. Any and all new data on the outcomes for bone healing associated with orthobiologic factors administered alone, or better, in combination could be added to the training dataset. A larger dataset would improve the ability of an ANN to identify combinations of factors with the potential to outperform single agents in promoting bone healing.

## Supporting information

**S1 Text. Setting up the training data.**
(PDF)

**S2 Text. Deep neural networks.**
(PDF)

**S3 Text. Finding the right neural network.**
(PDF)

**S4 Text. Possible implications of the character of the best network.**
(PDF)

**S1 Dataset. Complete set of training data used in machine learning.**
(XLSX)

## Acknowledgments

This work was conducted in the absence of any public or private funding.

## Author Contributions

**Conceptualization:** Albert T. Anastasio, Thomas J. Anastasio.

**Data curation:** Albert T. Anastasio.

**Formal analysis:** Albert T. Anastasio, Bailey S. Zinger, Thomas J. Anastasio.

**Investigation:** Albert T. Anastasio.

**Methodology:** Albert T. Anastasio, Bailey S. Zinger, Thomas J. Anastasio.

**Project administration:** Thomas J. Anastasio.

**Resources:** Albert T. Anastasio.

**Software:** Bailey S. Zinger, Thomas J. Anastasio.

**Supervision:** Thomas J. Anastasio.

**Visualization:** Bailey S. Zinger, Thomas J. Anastasio.

**Writing – original draft:** Albert T. Anastasio, Thomas J. Anastasio.

**Writing – review & editing:** Albert T. Anastasio, Thomas J. Anastasio.

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
