## [Decision Letter · Decision Letter 0]

8 Aug 2022

PONE-D-22-16681A Novel Application of Neural Networks to Identify Potentially Effective Combinations of Biologic Factors for Enhancement of Bone Fusion/RepairPLOS ONE

Dear Dr. Anastasio,

Thank you for submitting your manuscript to PLOS ONE. After careful consideration, we feel that it has merit but does not fully meet PLOS ONE’s publication criteria as it currently stands. Therefore, we invite you to submit a revised version of the manuscript that addresses the points raised during the review process.

ACADEMIC EDITOR: Please revise the manuscript based on the reviewers suggestions and comments.  

We look forward to receiving your revised manuscript.

Kind regards,

Kathiravan Srinivasan

Academic Editor

PLOS ONE

Journal Requirements:

Reviewers' comments:

Reviewer's Responses to Questions

**Comments to the Author**

1. Is the manuscript technically sound, and do the data support the conclusions?

Reviewer #1: Yes

Reviewer #2: Yes

2. Has the statistical analysis been performed appropriately and rigorously? 

Reviewer #1: Yes

Reviewer #2: Yes

3. Have the authors made all data underlying the findings in their manuscript fully available?

Reviewer #1: Yes

Reviewer #2: Yes

4. Is the manuscript presented in an intelligible fashion and written in standard English?

Reviewer #1: Yes

Reviewer #2: Yes

5. Review Comments to the Author

Reviewer #1: Thank you very much for the chance to review this manuscript. The authors extracted literature data on bone healing to train DL models for prediction modeling with regard to treatment success. I read the manuscript with great interest and think it provides an innovative way of processing the current data patterns in the literature to improve patient treatment outcomes. Although the available data points on this topic are small at the moment and the provided models’ capacities are insufficient for accurate predictions, I think this technique could be used in the future when more data is available in literature or other fields where sufficient data is already available. I recommend publishing this work as I believe it significantly contributes to the literature. I only have some minor comments which I recommend addressing before publication:

1)how was the literature search performed? The process of data collection (literature search) should be provided in more detail (e.g., search algorithm, which databases), as the results are based on them.

2)P6 “The analysis indicates that combinations of BMP2, BMP7, and OG, perhaps including ES or BMAC, each at the high end of their ranges as reported in relevant studies, should outperform combinations that lack those components. “

Which relevant studies? They should be cited after this statement.

3)please provide the hardware and software used for model building and training for reproducibility of methods provided

4)usually, there is also a validation dataset used that the algorithms have not seen before, resulting in a train, test, and validation dataset. As the testing data will also be used in the hyperparameter tuning process (to evaluate the best parameters), it is recommended by available guidelines on the reporting of machine and deep learning algorithms to include this type of internal validation. However, I see that with the limited data points, this seems to be not feasible. I recommend discussing this point in 1-2 sentences in the discussion section.

Reviewer #2: i. In abstract add more regarding the novel contribution of the manuscript.

ii. Literature review is shallow in context of various modeling technics.

iii. For Data Driven AI model specially Deep Learning Model needs large data, tyr to research and add more.

iv. Novelty of the work in terms of comparison to existing comparable models is missing,

you can refere article for refrence,Automated skull damage detection from assembled skull model

using computer vision and machine learning.Int. j. inf. tecnol. 13, 1785–1790 (2021). https://doi.org/10.1007/s41870-021-00752-5

v.The running time of the proposed model is not counted in the experiment. Time complexity is an important performance index.

6. PLOS authors have the option to publish the peer review history of their article (what does this mean?). If published, this will include your full peer review and any attached files.

Reviewer #1: No

Reviewer #2: **Yes: **Amol Mangrulkar

---

## [Author Response · Author response to Decision Letter 0]

21 Sep 2022

Reviewer #1: Thank you very much for the chance to review this manuscript. The authors extracted literature data on bone healing to train DL models for prediction modeling with regard to treatment success. I read the manuscript with great interest and think it provides an innovative way of processing the current data patterns in the literature to improve patient treatment outcomes. Although the available data points on this topic are small at the moment and the provided models’ capacities are insufficient for accurate predictions, I think this technique could be used in the future when more data is available in literature or other fields where sufficient data is already available. I recommend publishing this work as I believe it significantly contributes to the literature. I only have some minor comments which I recommend addressing before publication:

Authors: We thank the reviewer for their positive and constructive comments, and we appreciate the reviewer's recognition of our purpose in providing this initial analysis as a framework for incorporating larger quantities of experimental data as they become available. Addressing the reviewer’s concerns has improved the clarity and thoroughness of our manuscript. 

1) How was the literature search performed? The process of data collection (literature search) should be provided in more detail (e.g., search algorithm, which databases), as the results are based on them.

A1.1) We searched the MEDLINE database using the PubMed search engine. We used specific orthobiologic factors as search terms. Specifics on the search are given in detail in the description of Step 1 in the Methods section of the revised manuscript. 

2)P6 “The analysis indicates that combinations of BMP2, BMP7, and OG, perhaps including ES or BMAC, each at the high end of their ranges as reported in relevant studies, should outperform combinations that lack those components.” Which relevant studies? They should be cited after this statement.

A1.2) This issue is confused because the sentence is misleading as written. It has been rewritten as: "The analysis indicates that combinations of BMP2 (15mg), BMP7 (5mg), and OG (150ug), each at the high end of their ranges as shown in the input/desired-output table, and perhaps including ES or BMAC (see Supplementary Text S1 and the Supplementary Spreadsheet), should outperform combinations that lack those components." The Supplementary Spreadsheet is an Excel spreadsheet that has every input/desired-output pair used in this analysis, with quantification in appropriate units as reported in the literature. Supplementary Text S1 provides details on how each input and desired output were measured, and also provides a complete list of the published articles from which all the input/desired-output data were drawn. References to the supplementary materials have been clarified throughout the revised text.

3) Please provide the hardware and software used for model building and training for reproducibility of methods provided

A1.3) All computer programs were written in-house in MATLAB™ and run on Intel Core-i5 based Dell™ workstations. Algorithms were developed directly from their descriptions in the primary literature, as summarized and cited in Supplementary Text S2. The core set of programs and the entire dataset are available for free download at https://github.com/SigmoidNetmaker/deep-network-orthobiologics. Statements of these facts have been added to the revised version at the end of the first paragraph of Methods. 

4) Usually, there is also a validation dataset used that the algorithms have not seen before, resulting in a train, test, and validation dataset. As the testing data will also be used in the hyperparameter tuning process (to evaluate the best parameters), it is recommended by available guidelines on the reporting of machine and deep learning algorithms to include this type of internal validation. However, I see that with the limited data points, this seems to be not feasible. I recommend discussing this point in 1-2 sentences in the discussion section.

A1.4) We did divide the input/desired-output data into training and testing sets. In assessing the ability of each network type to generalize, we divided the set of 225 input/desired-output patterns into 175 training patterns and 50 testing patterns. We computed the root mean square (RMS) error over the 50 testing patterns after training on the 175 training patterns. We then retrained each network type 10 times, choosing a different training set and testing set at random each time. The best generalizing network type was the one having the lowest average RMS error over the 10 retrainings (see also Supplementary Texts S3 and S4). The full set of 225 input/desired-output patterns was used to train the best generalizing ANN type. This explanation has been added to the descriptions of Step 3, Step 4, and Step 5 in the Methods section of the revised manuscript. 

Reviewer #2: i. In abstract add more regarding the novel contribution of the manuscript.

A2.1) We thank the reviewer for acknowledgement of the novelty of our contribution. Indeed, PubMed and GoogleScholar searches of the standard keywords “orthobiolologics” <AND> “machine learning” yield no matches (except that GoogleScholar finds our medRxiv preprint describing this work). Our use of machine learning in this context is unprecedented. This is now noted in the Abstract of the revised version. This and the reviewer’s other comments have helped us improve our manuscript. 

ii. Literature review is shallow in context of various modeling technics.

A2.ii) The field of Machine Learning (ML) is vast. Still, it is only a small part of the statistical modeling field more generally. A deep review of ML, not to mention of statistical modeling more generally, is well beyond the scope of our article, whose purpose is to provide a framework for leveraging experimental data on bone healing to identify potentially new treatment combinations for use as adjuvants in orthopedic surgery. Our purpose is stated more clearly in the Introduction of the revised manuscript. 

iii. For Data Driven AI model specially Deep Learning Model needs large data, tyr to research and add more.

A2.iii) Large datasets are indeed preferable when using machine learning to train ANN AIs, but a large dataset is simply not available in this case. For training our ANNs, we used all the experimental orthobiologics data we could find. We state this fact in the Methods section of the revised manuscript. Our intention is to provide a computational framework to which further data can be incorporated as they become available. We address these issues and limitations in the Introduction and Discussion.

iv. Novelty of the work in terms of comparison to existing comparable models is missing,you can refere article for refrence,Automated skull damage detection from assembled skull model using computer vision and machine learning.Int. j. inf. tecnol. 13, 1785–1790 (2021). https://doi.org/10.1007/s41870-021-00752-5

A2.iv) This is the first time that ML and ANNs have been applied in the extraction of knowledge from a set of experimental observations on bone healing, and use of that knowledge to identify potentially effective combinations of biologic factors as adjuvants in orthopedic surgery. This is now mentioned in the Abstract and Introduction of the revised manuscript. As in many other areas of biomedicine, the application of ML and AI has grown rapidly in orthopedics because they often outperform older methods (eg statistical or biophysically based). Several extensive reviews specifically on ML in orthopedics have appeared recently. These are summarized and cited in the Introduction of the revised manuscript. The reviewed studies employed not only ANNs and ML but many other statistical modeling modalities as well; their description is outside the scope of our manuscript. Knowledge was extracted most often from radiological images (eg X-radiology, computed tomography, magnetic resonance, bone density, etc) but also from patient records, gate patterns, various sensors (eg force sensors, kinetic skeletal trackers, video, wearable device records, etc), and other kinds of measurements (eg biochemical, biomechanical, and bioelectrical). The knowledge extracted by the AI was then used for diagnosis of such disorders as osteo arthritis, ligament deficiency, cartilage lesion, rotator cuff pathology, spinal stenosis, scoliosis, and detection of weakness, age, fracture, tumor, and other bone abnormalities. The AI also can be used to predict surgical outcomes including survival, risk, complications, morbidity, mortality, treatment cost, duration as inpatient, and even patient satisfaction. In orthopedic surgery, AIs can be used to detect and identify surgical landmarks or targets such as specific bones, ligaments, and other skeletal components. None of the reviews, nor our own PubMed and Google Scholar searches, identifies anything like the framework we describe in our manuscript for using experimental data on bone healing to identify potentially effective combinations of biologic factors as adjuncts in orthopedic surgery. Our framework is based on standard, canonical ANNs and ML, which renders transparent the application of these forms of AI to orthobiological data. This is also stated in the Introduction of the revised manuscript. 

v.The running time of the proposed model is not counted in the experiment. Time complexity is an important performance index.

A2.v) All ANNs were trained for 50,000 training iterations using backpropagation (and closely related algorithms), where each iteration constituted one input/desired-output presentation (see Supplementary Text S3). Algorithms were written in MATLAB™ and run on Intel Core-i5 based Dell™ workstations. Training for 50,000 iterations took about 15 minutes. Times for blocks of multiple trainings are based on the 15-minute time for 50,000 iterations. Approximate computing times are now provided in the descriptions of Step 3, Step 4, and Step 5 and in the initial paragraph of the Methods section of the revised manuscript. A fuller account of the time required for all of the ANN and ML optimizations we undertook is given in Text S3 in the revised version of the Supplement. Indeed, optimizing the main ML parameters for all 16 ANN types was a major undertaking and required about 1 month of computer time, not including autoencoder optimization and preliminary runs. The routines were run on dedicated lab machines so the long processing time did not interfere with other work in the lab.

---

## [Decision Letter · Decision Letter 1]

10 Oct 2022

A Novel Application of Neural Networks to Identify Potentially Effective Combinations of Biologic Factors for Enhancement of Bone Fusion/Repair

PONE-D-22-16681R1

Dear Dr. Anastasio,

We’re pleased to inform you that your manuscript has been judged scientifically suitable for publication and will be formally accepted for publication once it meets all outstanding technical requirements.

Kind regards,

Kathiravan Srinivasan

Academic Editor

PLOS ONE

Additional Editor Comments (optional):

Reviewers' comments:

Reviewer's Responses to Questions

**Comments to the Author**

1. If the authors have adequately addressed your comments raised in a previous round of review and you feel that this manuscript is now acceptable for publication, you may indicate that here to bypass the “Comments to the Author” section, enter your conflict of interest statement in the “Confidential to Editor” section, and submit your "Accept" recommendation.

Reviewer #1: All comments have been addressed

Reviewer #2: All comments have been addressed

2. Is the manuscript technically sound, and do the data support the conclusions?

Reviewer #1: Yes

Reviewer #2: Yes

3. Has the statistical analysis been performed appropriately and rigorously? 

Reviewer #1: Yes

Reviewer #2: Yes

4. Have the authors made all data underlying the findings in their manuscript fully available?

Reviewer #1: Yes

Reviewer #2: Yes

5. Is the manuscript presented in an intelligible fashion and written in standard English?

Reviewer #1: Yes

Reviewer #2: Yes

6. Review Comments to the Author

Reviewer #1: The authors have sufficiently addressed all comments. The can be published in the current version from my side.

Reviewer #2: The Author adress coment no1 which In abstract add more regarding the novel contribution of the manuscript

and add the novelty of thier work in abstarct which i verifies and all others comments address in all respect , Now no futher comments my side.

7. PLOS authors have the option to publish the peer review history of their article (what does this mean?). If published, this will include your full peer review and any attached files.

Reviewer #1: No

Reviewer #2: **Yes: **DR AMOL L. MANGRULKAR

---

## [Editor Report · Acceptance letter]

20 Oct 2022

PONE-D-22-16681R1 

A Novel Application of Neural Networks to Identify Potentially Effective Combinations of Biologic Factors for Enhancement of Bone Fusion/Repair 

Dear Dr. Anastasio:

I'm pleased to inform you that your manuscript has been deemed suitable for publication in PLOS ONE. Congratulations! Your manuscript is now with our production department. 

Kind regards, 

on behalf of

Dr. Kathiravan Srinivasan 

Academic Editor

PLOS ONE